# Cervical Imaging in the Low Resource Setting: A Review

**DOI:** 10.3390/bios12100786

**Published:** 2022-09-24

**Authors:** Mariacarla Gonzalez, Tananant Boonya-Ananta, Purnima Madhivanan, Jessica C. Ramella-Roman

**Affiliations:** 1Biomedical Engineering Department, Florida International University, 10555 W Flagler St., Miami, FL 33174, USA; 2Department of Health Promotion Sciences, Mel and Enid Zuckerman College of Public Health, University of Arizona, Tucson, AZ 85724, USA; 3Public Health Research Institute of India, Mysore 560020, India; 4Herbert Wertheim College of Medicine, Florida International University, Miami, FL 33174, USA

**Keywords:** cervical cancer, low resource setting, cervical imaging devices

## Abstract

Cervical cancer is one of the most significant global health inequities of our time and is the fourth most common cancer in women worldwide, disproportionally affecting developing countries where the disease burden is 84%. Sometimes referred to as preventable cancer, it progresses slowly, providing a window of time for routine screening in which pre-cancerous lesions can be identified and treated. The high mortality rate can be attributed to many reasons, including the high cost of cytology-based screening, lack of human resources to conduct screening, and inadequate preventive medicine services and systems. Due to its slow progression, early intervention is feasible with appropriate screening. However, the standard screening procedures require access to lab-based tests and physician expertise. Several imaging devices have been introduced in the literature to aid cervical screening in low-resource settings. This review details the instrumentation and clinical testing of devices currently deployed in low-resource locations worldwide. The devices’ imaging, portability, illumination, and power requirements (among other metrics) are documented with specifics of human pilot studies conducted with these tools.

## 1. Introduction

Cervical cancer is the fourth most common cancer in women. More than half a million women are diagnosed yearly due to persistent human papillomavirus (HPV) infection, with mortality as high as 311,000 [1]. According to the World Cancer Research Fund, developing countries have 84% of the global disease burden and 80% of the mortality due to a lack of effective screening programs [2]. This causes cervical cancer to be an example of global health inequity since the slow-progressing disease provides time for detecting and treating pre-cancerous lesions. Many women in low and middle-income countries (LMICs) seek clinical care once they experience persistent cancer symptoms. In contrast, cervical cancer screening programs in high-income countries have helped reduce mortality significantly [3]. Several screening techniques have been developed and implemented to aid in low-resource cervical screening.

Literature reviews describing the optical modalities available for cervical cancer detection have been introduced by Novikova, Hill et al. and Olpin et al. and others where they have covered modalities, such as ultrasound, optical coherence tomography, and spectroscopy (among others), as well as their clinical outcomes [4]. Softland et al. looked at two handheld colposcopes, the Gynocular and the Enhanced Visual Assessment (EVA) System by Mobile ODT, and compared their capabilities for the use in female genital schistosomiasis [5].

This manuscript reviews uniquely cervical imaging devices for deployment in low-resource settings that can potentially be implemented for cervical cancer screening and diagnosis.

### 1.1. Anatomy of The Cervix

The cervix is a cylindrical structure that connects the vaginal canal (ectocervix) to the uterus (endocervix). It is 2–3 cm long, composed mainly of epithelium and stroma. There are two main types of epithelia present in the cervix: columnar and stratified squamous. The columnar epithelium is the lining found in the endocervix and secretes mucus. The stratified squamous epithelium is located in the ectocervix and is a continuation of the vaginal epithelium. The location where these two epithelia meet is called the squamocolumnar junction (SCJ). The location of the SCJ varies depending on continuous cervical remodeling, the main factors being age and hormones (e.g., the SCJ is found in the external os in younger women) [6,7]. The cervix contains a thick stroma layer under both epithelial types, mainly muscular, elastic, and fibrous tissues. The fibrous stroma occupies three areas with unique orientations surrounding the cervical canal. The inner canal and outer cervix are composed of longitudinally aligned collagen, and in between can be found circumferentially aligned collagen [8]. Figure 1 illustrates an anatomical representation of the cervix.

HPV infection is the principal cause of cervical cancer. Types 16 and 18 are responsible for 71% of cases; however, when including HPV types 4, 11, 16, 18, 31, 33, 45, 52, and 58, the percentage of cervical cancer cases due to HPV rises to 90%. HPV is a family of DNA viruses (approximately 15 that can infect the genital tract) that target basal epithelial cells and cause benign and malignant lesions [9,10]. Common cancers include squamous cells, adenocarcinomas, sarcomas, and small cell neuroendocrine tumors. The immune system clears most infections; if not, the virus proceeds to attack the cells in the cervical SCJ [4,10,11]. Persistent infection can spread and break through the basal membrane to become an invasive cancer [4].

### 1.2. Disease Progression

During disease progression, three types of neoplastic states affect the epithelium. Cervical intraepithelial neoplasia (CIN) of first grade or CIN 1, affects one-third of the epithelium and is considered mild and likely caused by a transient HPV infection, which should clear naturally. CIN 2, which affects two-thirds of the epithelium, is a moderate case and combination of self-clearing and pre-cancerous lesions. CIN 3 is considered severe as it affects the whole epithelium; it is a pre-cancer state since the lesions are unlikely to clear up naturally. Using the Bethesda System (classification system used for cytological diagnosis and treatment decisions), CIN 1 is a low-grade squamous intraepithelial lesion (LSIL) and CIN 2 and 3 are high-grade squamous intraepithelial lesions (HSIL). Invasive cervical cancer is a slowly progressing disease, generally taking more than ten years to fully develop from infection [4,12,13].

Cervical neoplasia is related to changes in both the stroma and epithelial cells [14,15,16,17]. Stromal changes stimulate and precede neoplastic progression. Moreover, carcinogenesis results from defective communication between the epithelium and the stroma [15,17]. The extracellular matrix (ECM) can regulate growth, death, gene expression, and migration, among other processes, all of which regulate physiologic processes such as angiogenesis, tissue morphogenesis, embryonic development, and pathological processes. Furthermore, stroma and tumor cells can exchange growth factors for activating neighboring ECM and aiding the expansion of neoplastic cells [14]. The deregulation between the stroma and the epithelium communication promotes carcinogenesis [15,16]. Neoplastic progression results in changes to the stroma and, therefore, the collagen matrix, which leads to changes in stromal scattering and can be used for optical contrast in the diagnostic measurement of neoplastic tissues [17].

### 1.3. Cervical Testing and Treatment

The standard procedure for cervical cancer diagnosis in the United States includes liquid-based cytology (Pap test) and DNA testing for high-risk HPV. Colposcopy, biopsy, and histological confirmation are performed if abnormal results are obtained. This procedure, however, requires a high level of quality standards, such as trained personnel, medical coverage, and follow-up visits. Therefore, the World Health Organization (WHO) recommends a screen and treat approach, where the primary screening test should be HPV DNA detection every five to ten years after the age of 30 [18]. Due to previous recommendations, current screening practices include HPV testing, visual inspection with acetic acid (VIA), and cytology, all followed by treatment. Another commonly used screening option is visual inspection with Lugol’s iodine (VILI), although not explicitly recommended by the WHO. Some of the practices mentioned above cannot be used in the general population; for example, VIA testing is not appropriate for women older than 50 since the transformation zone (where the lesions usually start) moves into the endocervical canal after menopause. The choice of screening techniques depends highly on the local resources, although the latest recommendations by WHO highly recommend the switch from previously mentioned methods to HPV DNA screening due to the objectivity of the test [18].

#### 1.3.1. HPV DNA Testing, Cytology, Colposcopy, and Biopsy

Cervical cancer screening in the United States consists of multiple stages. HPV DNA co-testing and cytology (or Pap smear) are the first steps for every cervical cancer diagnosis. A speculum is inserted into the vaginal canal to collect cells from the cervix. The cells are analyzed for abnormality and apparent changes. Cytology results are difficult to score as it has been shown that there is low interobserver agreement. Stoler et al. found only 47.1% agreement in interpreting HSIL for cytology results when comparing the original diagnosis with a quality control group [19]. HPV DNA testing determines the presence of high-risk HPV with specificity and accuracy of 55.6% and 75.8%, respectively, and a positive predictive value of 84.8% [20].

A second step in the cervical screening is colposcopy when abnormal cells are found (i.e., positive Pap smear). Colposcopy is a visual inspection conducted by trained physicians with a colposcope (a clinical microscope with 3–15 times magnification) that allows for a closer look at the uterine cervix. The accuracy of this procedure is highly dependent on clinicians’ training level and experience. The diagnostic value of the technique has been reported to have high sensitivity (85%). Still, low specificity (69%), meaning the abnormal location can be found, but the severity of the lesion is often inaccurate [21,22,23,24].

Furthermore, the interobserver variability for colposcopic data has a kappa value of 0.40 [25]. As part of the colposcopy, a biopsy is usually performed where a small portion of the cervix is sampled. Similar to cytology, biopsies have a low interobserver agreement. A study on 2237 cervical histologies showed that the agreement between the original pathologist and the quality control group overlapped only 42.7% of the time for CIN1 cases [19].

#### 1.3.2. Visual Inspection

VIA involves applying a 3–5% acetic acid solution to the ectocervix. This application will turn abnormal cells in the epithelium to an opaque white color (referred to as acetowhite), and the tissue is considered VIA positive. These acetowhite lesions are due to the coagulation of proteins in the cells with acetic acid since neoplastic tissue will have a higher protein content than normal tissue. The positive predictive value of VIA is 16.7%, and the negative predictive value of 99%. The specificity and sensitivity are 79.4% and 71.8%, respectively [26,27]. These results translate to many false positives leading to overdiagnosis and overtreatment.

Another visual inspection technique, VILI, involves applying Lugol’s iodine to the cervical epithelium. This solution reacts with glycogen in normal healthy tissue and turns black upon exposure. In the presence of neoplastic tissue, the glycogen is reduced or absent, and the solution turns the epithelium yellow. The positive predictive value of VILI is 16.8%, and the negative predictive value of 99.7%, resulting in many false positives. The specificity and sensitivity of VILI are 86% and 88%, respectively [27,28].

Visual inspection for cervical screening suffers from low reproducibility and results in variation depending on the subjectivity of the interpretation of the results [29]. It has also been shown that age, parity, menopause, and HPV presence can influence the outcome of visual inspection tests and the level of training of the healthcare providers [30]. However, the low cost and real-time results from visual inspection tests make it ideal for the low resource settings and the screen-and-treat approach, especially in areas of high cervical cancer incidence and low medical resources [28,29,31]. To overcome the current screening issues using VIA and VILI, better training of healthcare personnel is needed. Moreover, Raifu et al. recommend specifically better training of personnel on the definition and interpretation of acetowhite lesions of the cervical epithelium in these settings [30].

#### 1.3.3. Treatments

The treatments recommended for cervical neoplasia are directed at removing or destroying the transformation zone and abnormal areas found in the cervix. Two main treatment routes include ablation and excision (although there is ongoing research for alternate treatments) [32]. Using ablative treatment, the abnormal tissue is destroyed by heating through thermal coagulation or freezing it via cryotherapy. The excisional route removes tissue by large loop excision of the transformation zone (LLETZ) or by cold knife cone (CKC), also known as conization of the cervix [18].

### 1.4. HPV Vaccines

There have been three HPV vaccines available since 2006, although only one is currently used in the United States. Gardasil 9 is a 9-valent vaccine that targets HPV types 6, 11, 16, 18, 31, 33, 45, 52 and 58. The vaccine has an efficacy close to 100% for young adolescents 9–15 years old [33]. The vaccine targets infections in anatomical areas other than the cervix (e.g., vulva, penis, anus). Although HPV vaccination has reduced the number of infections in women since its introduction, it does not cover all 15 high-risk HPV types [34]. Moreover, it is expensive and difficult to implement in developing countries, leaving screening and treatment of precancerous lesions as the main preventive methods [35].

The slow progression of cervical cancer, the anatomic accessibility, and the possible treatment of precancerous lesions make early screening an effective management [4,12]. Due to the high costs of traditional cervical screening procedures, several devices have been developed to increase access to cervical testing in the low-resource setting. This review paper introduces current cervical imaging devices designed for deployment in the low resource setting, their specifications, and clinical outcome.

## 2. Limits of Cervical Screening in the Low Resource Setting

Limitations on cervical screening in low-resource settings include an array of reasons. Common issues include the lack of regular participation in patient screening due to social and cultural taboos, health literacy, inadequate sampling and management of smears by clinicians, interpretation errors from pathologists, and lack of screening programs that can reach target populations [36]. Operational limitations to existing screening tests, such as cytology, VIA/VILI, and HPV DNA-based tests, include a lack of trained workforce, timeliness of test result availability, the possibility of overtreatment, and the need for laboratory setup, among others [37,38]. Moreover, screening with colposcopes is challenging to implement since they are costly, electricity dependent, and need high maintenance [39]. They are also voluminous and heavy, making them difficult to transport outside a clinical setting.

The World Health Organization has previously noted that even a once-in-a-lifetime Pap smear screening can significantly reduce the incidence and mortality of cervical cancer [37]. Introducing portable, low-cost devices aims to close this gap in screening limitations.

## 3. Cervical Imaging Targeted for Neoplastic Detection

### 3.1. Callascope

#### 3.1.1. Device

The Callascope is a speculum-free device used for capturing images of the cervix [40,41,42,43]. The Callascope was developed at the Department of Biomedical Engineering at Duke University, Durham, NC, USA. The Callascope is designed to create a speculum-free imaging system composed of an introducer and a slender camera. The introducer is a Calla Lily-shaped silicone hollow tube which can be inserted into the vagina (Figure 2) [40]. The introducer is approximately 30 mm at the larger proximal end and 12 mm at the distal end [41].

The asymmetric tip is designed to allow rotation of the introducer to tilt the cervix into a favorable viewing position. The light source is composed of a ring illuminator with four white LEDs. The camera and housing can be inserted into the introducer to be positioned for imaging the cervix. The camera body is a slim 9 mm diameter tube with a length of approximately 120 mm. The camera is a 2 to 5 Megapixel CMOS sensor with a lens [40,41]. The camera is fitted with a hydrophobic window at the tip and is positioned in the center of the ring illuminator. The camera is set to a working distance of 25 to 30 mm from the cervix when inserted into the introducer. The Callascope has a field of view of 35 mm. At a working distance of 30 mm and 4× magnification, the smallest resolved feature on a USAF 1951 resolution target was 99.2 µm.

#### 3.1.2. Clinical Testing

Clinic testing of the device has been performed in both the United States and Ghana, looking at two different environments of the Callascope: clinician usage and self-conducted imaging of the cervix [40]. Participant eligibility included healthy females 18 years or older. The number of participants in Ghana comprised 25 for clinician testing and 10 for individual usage. In the U.S., 28 participants for clinicians, and 12 were for self-imaging. Participants underwent a pre-exam survey to document demographical information and perceptions using a speculum, Callascope, and clinician vs. self-examination. Post-examination survey was conducted using a modified Universal Pain Assessment tool alongside a written description. Image quality was assessed using one point for visualization of the os and one for each of the four cervical quadrants.

The overall assessment shows a higher preference for the Callascope vs. a standard speculum above 75% in both testing sites (CITE SR 2020). In studies performed by clinicians, the Callascope enabled visualization of the os for 78.6% of U.S. and 80% of Ghana participants. The speculum-based imaging shows the visualization of the os for 96% in the U.S. and 100% in Ghana. Table 1 assesses cervical quadrant visualization for clinician usage [40]. Over 60% of participants in both sites found the Callascope easy to insert and use for self-imaging. No patients indicated extreme discomfort, and over 70% of participants stated no or slight pain in the post-examination survey.

### 3.2. High-Resolution Microendoscope (HRME)

#### 3.2.1. Device

The high-resolution microendoscope (HRME) is a fluorescence optical imaging system employed for cervical cancer screening developed at Rice University, Houston, TX, USA. The system light source consists of a 455 nm LED coupled with a fiber bundle. This wavelength is used to excite proflavin, an FDA-approved fluorescent DNA label used to dye nuclei from the cytoplasm of cells. A topical solution of proflavin is needed to be used along the HRME, where the fluorescent emission (510 nm) is captured with a CCD camera (also coupled to the fiber bundle) after passing through a 475 nm dichroic mirror. The probe, consisting of the fiber bundle, requires insertion through a speculum. To be in focus, probe contact with the cervical epithelium is needed. The HRME can provide real-time morphology and epithelial architecture with a field of view of 720 µm and a lateral resolution of 4 µm [44,45]. The device is portable and weighs 2.3 kg—although a new iteration has reduced the weight to 0.91 kg [46]. The HRME costs approximately $2450 mainly due to the computer tablet, although costs have been reduced with the introduction of a Raspberry Pi computer [47]. The device can be seen pictured in Figure 3.

#### 3.2.2. Clinical Testing

The HRME has been deployed in clinical settings such as Botswana, Brazil, the United States (Texas), and El Salvador [44,47,48,49,50,51]. A human study in Botswana was performed by first conducting a routine colposcopic examination. Then, a solution of proflavine hemisulfate was applied, and the HRME was inserted through a speculum to meet the cervix. Images were gathered for 26 patients from 52 sites; low-quality images were discarded. Calculating the average nuclear to cytoplasmic area ratio, a receiver operator characteristic (ROC) determined a specificity of 86% and a sensitivity of 87% high-grade neoplastic lesions (CIN2+) [44]. Another study in Brazil deployed the device in a colposcopy clinic in Barretos Cancer Hospital and a mobile diagnostic van that traveled to different communities. The portable device was used after routine colposcopy examination and application with proflavine solution. The study determined an average specificity and sensitivity of 48% and 92%, respectively, for identifying CIN2+ compared with histopathology [48]. The HRME system has also been used in oral and esophageal cancer diagnosis [52,53,54,55].

### 3.3. Snapshot Mueller Matrix Polarimeter

#### 3.3.1. Device

The snapshot Mueller matrix polarimeter is a portable optical imager introduced in 2020 by our group. The device is based on Mueller matrix polarimetric imaging and uses a ring illuminator to generate four different polarization states at 633 nm for the polarization state generator (PSG). Two Savart plates achieve the snapshot approach to develop four other rays with unique polarization information that are analyzed by a 45° polarizer—forming the polarization state analyzer (PSA)—and are detected on a CMOS camera. The polarimetric approach can provide quantitative information on the cervix using Mueller matrix decomposition since healthy (normal) and unhealthy cervixes behave differently to incident polarized light (especially the parameters of depolarization and retardance). The device’s field of view is 30 mm, allowing a full view of the cervix with a single snapshot. The device is noninvasive, although a speculum is needed to visualize the cervix. The cost of the device is approximately $2000 [56]. A picture of the device can be seen in Figure 4.

#### 3.3.2. Clinical Testing

The device was clinically deployed at the Public Health Research Institute of India (PHRII) in Mysore, India. Twenty-two patients were recruited, although six were excluded from the reported results due to image quality. The patients underwent cervical inspection as routine examination, and then the snapshot polarimeter was used to image the cervix. The results agreed with polarimetric imaging of healthy cervices, where there are high depolarization values for all patients. There was an exception for one patient diagnosed with a polyp, which showed lower depolarization values (as expected) [56,57].

### 3.4. Enhanced Visual Assessment (EVA) System

#### 3.4.1. Device

The Enhanced Visual Assessment (EVA) System developed by MobileODT (Tel Aviv, Israel) is a portable colposcope for enhanced analysis using VIA. The system utilizes a speculum to image the cervix. It can be used to augment the results from VIA by supplying the lighting and magnification needed as well as aiding the logging of images and information. The EVA system is portable, weighing 605 g, with a light source consisting of a white 3 W (3.6 V) LED. The battery-powered system can last up to ten hours of constant use. The system is equipped with a cellphone, with an optical zoom capability of 4x and a digital zoom capability of 16x. The onboard software provides real-time analysis capability and tracking for patient follow-up [58]. The EVA system utilizes an application to control the smartphone and a cloud-based image portal to store and view images [59]. The device cost is approximately $8200, including annual service and technical support. The device can be seen in Figure 5.

#### 3.4.2. Clinical Testing

Clinical testing of the EVA system was conducted at different sites. The device was used as primary screening co-testing along with cytology by the Fronteras Unidas Pro-Salud outreach program, which provided an early guide of suspicious areas in patients [60]. Another clinical study conducted in a hospital-based setting and an urban screening camp in Mumbai, India, showed an agreement between EVA and cytology in 157 cases out of 471 patients. Most disagreements in prognosis were due to misclassification of cervicitis in patients. It must be noted that EVA compared well against naked eye visualization in the screening camp, as well as collected information (such as age and socioeconomic status) that is often difficult to gather [61]. The device has also been included in protocols to screen HIV-infected women for cervical cancer in Rwanda.

Image quality was tested for images taken using the EVA system. A random subset of images in the MobileODT portal found that 73% of the images were of poor quality and could not be used further. To address this issue, an ongoing effort to determine the image quality in real-time is underway using machine learning methods [59].

### 3.5. Gynocular

#### 3.5.1. Device

The Gynocular is a small monocular colposcope developed by Gynius Plus AB, a company based in Stockholm, Sweden. The device functionality is like the colposcope but has the advantage of being pocket size and a total weight of 480 g. A self-holding speculum is used in conjunction with the device to access the cervix. The Gynocular offers an optical magnification of 5×, 8×, and 12× with a field of view ranging from 20 to 40 mm (depending on the magnification). The light source employs a 3 W/3.6 V warm white LED, and a green filter (530 nm) can be added to the imaging protocol. The battery on board can withstand at least two hours of use. The device is portable and can be used as a handheld device and mounted on a tripod for increased stability. A cellphone can also be coupled with a portable device to take images. The cost of the mobile device is approximately $3000. A picture of the Gynocular can be seen in Figure 6.

#### 3.5.2. Clinical Testing

The Gynocular has been compared to a standard colposcope in multiple clinical studies in Uganda, India, Bangladesh and Sweden [62,63,64,65,66,67,68]. One study tested VIA positive women in a clinical study in a hospital setting in Uganda. Sixty-seven women were included in this study, and visual scores given to the cervix state were 70.1% in agreement for both modalities, whereas 47 out of 67 measurements were in agreement [62]. Another clinical study performed in a colposcopy clinic in Bangladesh determined that the Gynocular had a sensitivity of 83.3% and a specificity of 23.6%, with a positive predictive value of 88.6% and a negative predictive value of 16.6% [64]. There was no significant difference between the Gynocular and the colposcope for identifying CIN2+ lesions in all clinical trials performed [62,64,65,66,68].

A summary table of the modalities and their specifications can be seen in Table 2.

### 3.6. Images

Sample images taken from each device discusses shown below. Images captured by the participants (self-imaging) of the cervix using the Callascope are shown in Figure 7. These images represent a subset of cervix data taken from 22 healthy volunteers to test the self-imaging abilities of the device.

Figure 8 shows images from Quinn et al. during a clinical study in Princess Marina Hospital in Botswana. The images on the left (A and D) are taken with a colposcope, where the white arrow signifies the area imaged with the HRME (B and E). The third row (C and F) is the histologic confirmation of the site probed. The top row pertains to a clinically normal region of the cervix, and the bottom row is from an abnormal part in the cervix [44].

Images from the snapshot Mueller matrix polarimeter can be observed in Figure 9. Three healthy human cervices are shown along with depolarization and retardance information, providing quantitative polarimetric details on the status of the tissue. These images were taken in a clinical pilot study in Mysore, India [56].

Mayoore et al. present a subset of images (as seen in Figure 10) taken by the EVA system showcasing different examples of image quality encountered in the MobileODT database. The figure shows representative images of levels of sharpness, going from low (very poor) to high (excellent) [59].

Figure 11, from Kallner et al., shows sample images taken with the Gynocular through a speculum imaging a normal HPV-positive cervix and an HPV-positive cervix with high-grade lesions [65].

## 4. Conclusions

We have described a set of tools for cervical imaging currently used in low-resource settings. The overall comparison between the specifications of all the devices, including illumination, power consumption, cost, and field of view (among other characteristics) have been analyzed and summarized (Table 2). The Callascope, the EVA, and the Gynocular work similarly to a colposcope providing images of the cervix to be examined by a physician, whereas the HRME and the snapshot Mueller matrix polarimeter provide more quantitative information via fluorescence and polarimetry, respectively. With the exception of the Callascope, the other four devices need the aid of a speculum to capture the cervical images. These devices range from $2000–$8200 and weight from 480–2300 g, allowing portability and field use. All devices have been clinically deployed in low-resource settings, where images have been collected for physician interpretation and quantitative assessment.

The limitations of cervical cancer testing in low-resource settings can range from cultural and social reasons to lack of screening programs, laboratory facilities, and electrical power availability. The introduction of cervical screening devices offering portability, low energy consumption, lower costs than traditional colposcopes, and the ability for widespread use enable developing and developed countries with remote and low-resource populations to receive cervical screening as preventive care. These devices are also being enhanced with machine learning algorithms to improve image quality and processing and aid in interpretation. The combination of the currently available technologies for cervical imaging as a screening tool with the addition of artificial intelligence will improve the testing outcome and reduce the effect of current limitations such as interpretation errors, test result timelines, and lack of workforce. The ability of a user to utilize the device without aid (as for the Callascope) is also seen as an asset.

## Figures and Tables

**Figure 1 biosensors-12-00786-f001:**
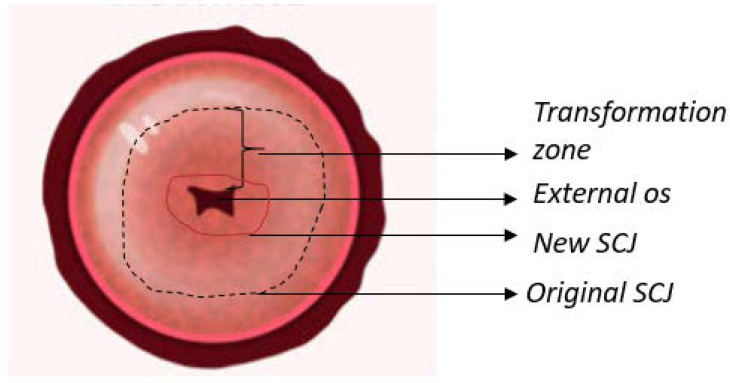
Anatomy of the uterine cervix highlighting the epithelium found on the surface, as well as the transformation zone and squamocolumnar junction (SCJ).

**Figure 2 biosensors-12-00786-f002:**
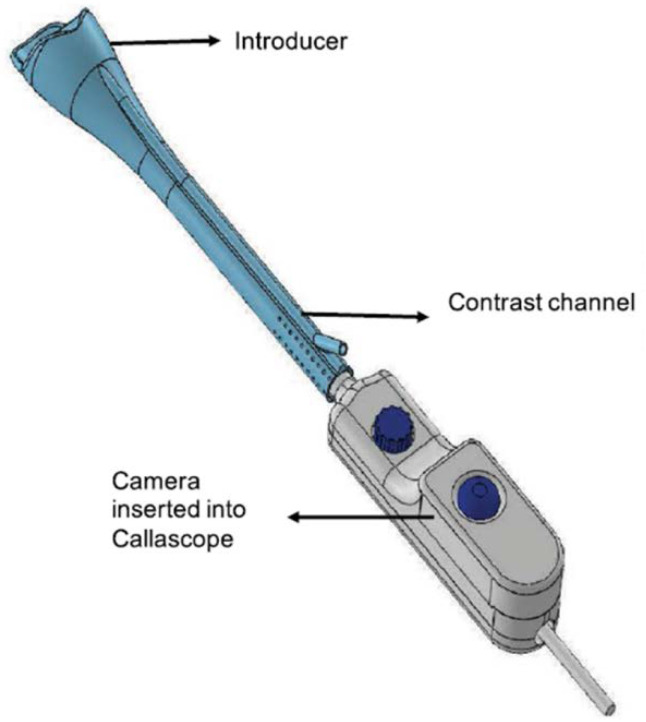
Schematic of the Callascope device, designed for self-insertion and aiming of the uterine cervix (Reprinted from Ref. [40]).

**Figure 3 biosensors-12-00786-f003:**
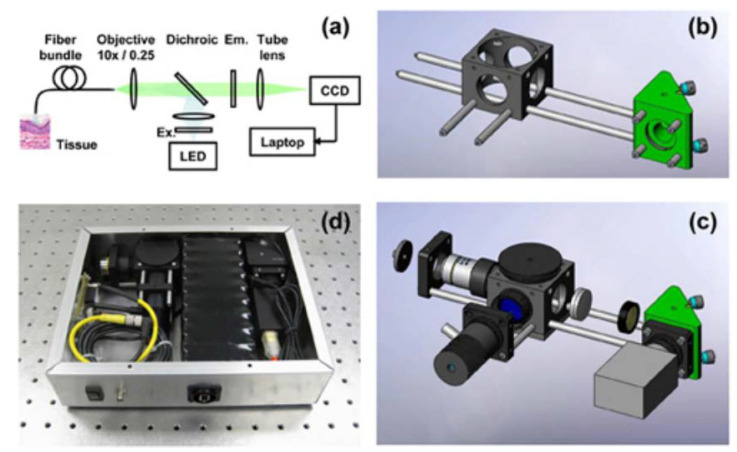
Schematic of HRME device used for fluorescence measurement of the cervical epithelium. (**a**) The optical schematic of the HRME system, (**b**) Opto-mechanical support structure assembly, (**c**) Opto-mechanical assembly with optical elements added, (**d**) Enclosed HRME system. (Reprinted with permission from Ref. [45]. Copyright 2022 MyJoVE Corporation).

**Figure 4 biosensors-12-00786-f004:**
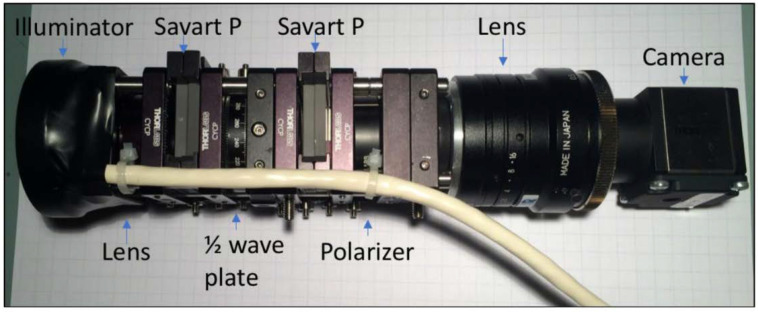
The snapshot Mueller matrix polarimeter used for polarization imaging of the uterine cervix (Reprinted from Ref. [56]).

**Figure 5 biosensors-12-00786-f005:**
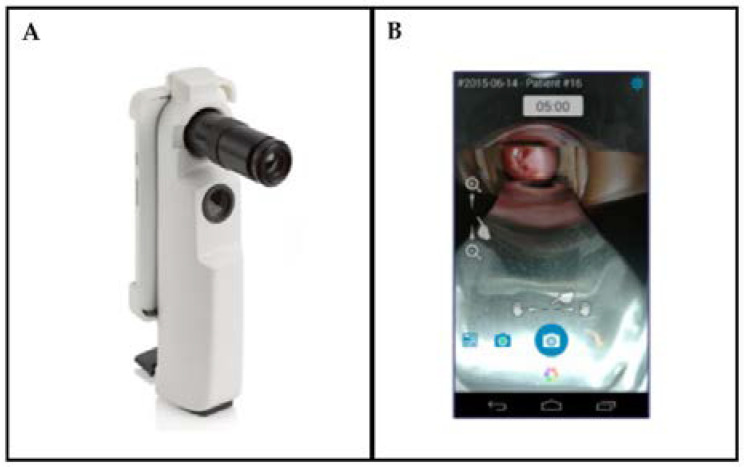
The EVA system (**A**) and an application image utilized to visualize the uterine cervix (**B**) (Reprinted with permission from Ref. [60]. Copyright 2016 Society of Photo-Optical Instrumentation Engineers (SPIE)).

**Figure 6 biosensors-12-00786-f006:**
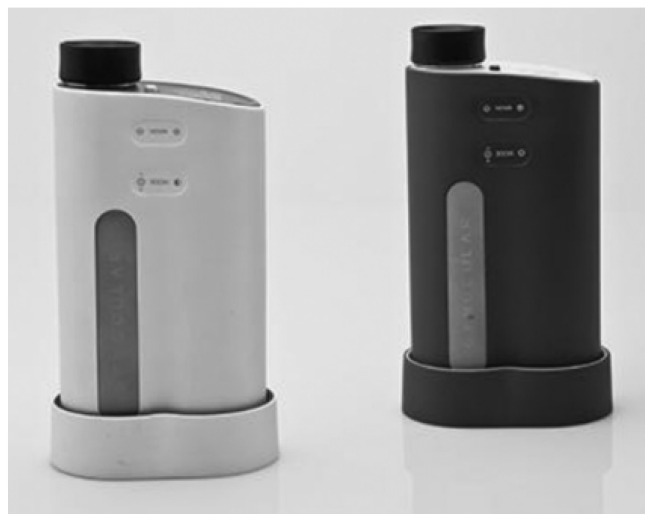
The portable colposcope, Gynocular, from Gynius Plus AB (Reprinted from [62]).

**Figure 7 biosensors-12-00786-f007:**
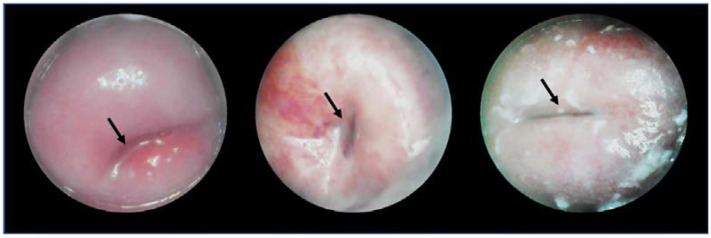
Sample images taken with the Callascope while self-imaging. The arrow points to the cervical external os (Reprinted from Ref. [40]).

**Figure 8 biosensors-12-00786-f008:**
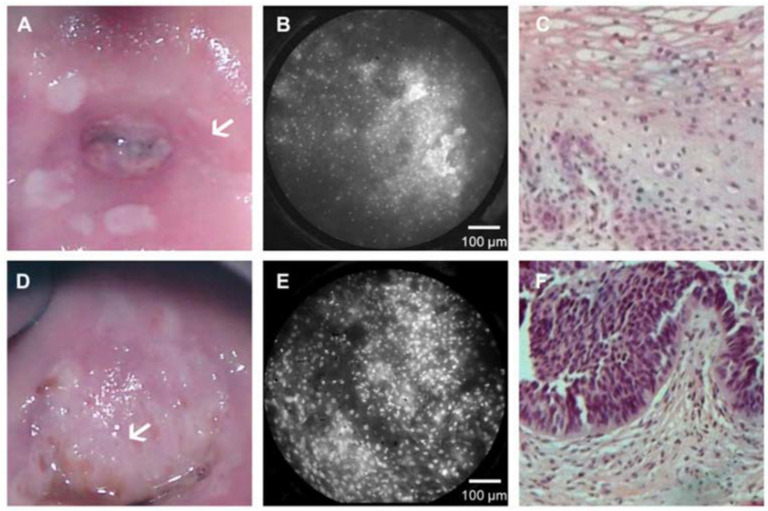
Representative cervix images of (**A**,**D**) the whole cervix, (**B**,**E**) the nuclei as seen by the HRME, and the (**C**,**F**) histopathology. The arrow points to the region of interest being imaged from A and D, shown in images B and E (Reprinted from Ref. [44]).

**Figure 9 biosensors-12-00786-f009:**
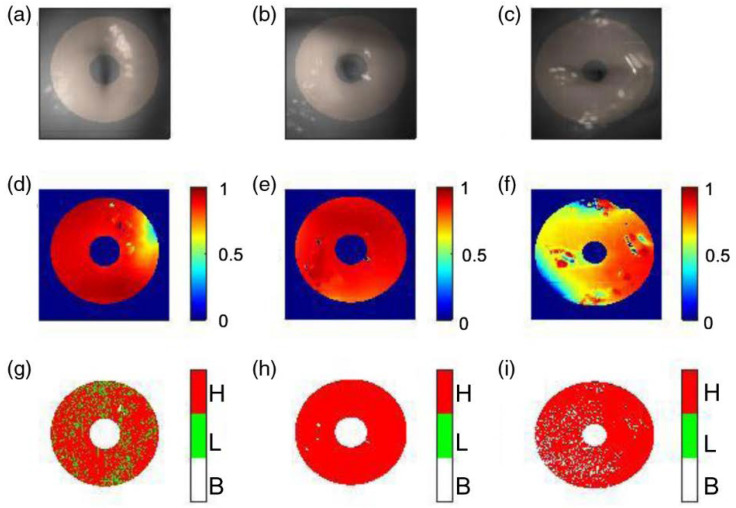
Cervical images taken by the snapshot Mueller matrix polarimeter, showing the (**a**–**c**) raw images and polarization information of (**d**–**f**) depolarization and (**g**–**i**) retardance (Reprinted from Ref. [56]).

**Figure 10 biosensors-12-00786-f010:**
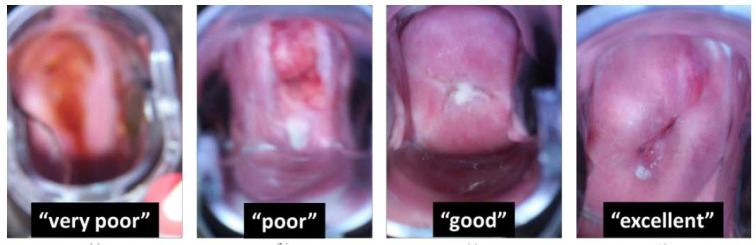
Example of images taken with the EVA system showing the different quality of images, ranging from very poor to excellent (Reprinted with permission from Ref. [59]. Copyright 2018 Society of Photo-Optical Instrumentation Engineers (SPIE)).

**Figure 11 biosensors-12-00786-f011:**
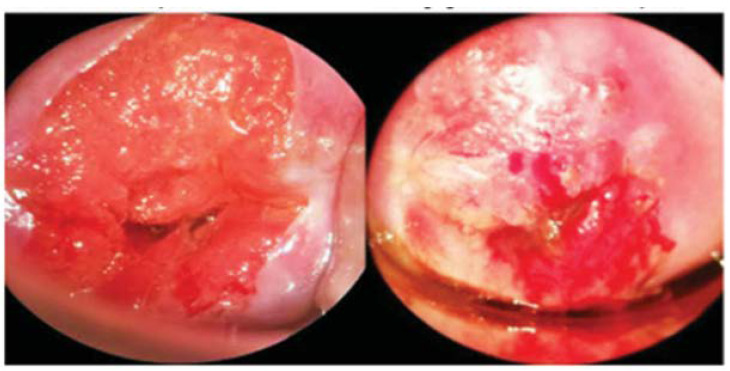
Images taken with the Gynocular, showing a normal cervix (**left**) and a cervix with high-grade lesions (**right**) (Reprinted with permission from Ref. [65]. Copyright 2015 Cambridge University Press).

**Table 1 biosensors-12-00786-t001:** Callascope cervical quadrant visualization.

View of at Least Cervical Quadrants	Callascope	
	U.S.	Ghana
	[%]	
2	89	84
3	72	71
4	50	44

**Table 2 biosensors-12-00786-t002:** Summary of the portable devices introduced for cervical imaging.

Device	Company	FOV	Weight	Power	Cost (Dollars)	Portable?	Magnification	Illumination	Can It Be Mounted?	Need Speculum?	Software Included?
Callascope	Duke University	30 mm	-	PC	-	Yes	4×	White ring LED	No	No	Yes
HRME	Rice University	720 microns	2.3 kg	PC	2450	Yes	10×	455 nm LED	No	Yes	Yes
snapshot Mueller matrix polarimeter	FIU	30 mm	-	PC	2000	Yes	none	(4) 633 nm LEDs	Yes	Yes	Yes
EVA	Mobile ODT	-	605 g	Battery	8200	Yes	4×, 16×	3 W (3.6 V) LED	Yes	Yes	Yes
Gynocular	Gynius	20–40 mm	480 g	Battery	3000	Yes	5×, 8×, 12×	3 W (3.6 V) LED	Yes	Yes	Yes

## Data Availability

Not applicable.

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
