# Peer review of "Cervical Imaging in the Low Resource Setting: A Review"

_biosensors, 2022, doi:10.3390/bios12100786_

Round 1

Reviewer 1 Report

Comments to the authors:

This manuscript details the instrumentation and clinical testing of devices currently deployed in low resource setting across the world. The imaging, portability, illumination and power requirements of devices are documented together with specifics of pilot human studies conducted with these tools. This is an interesting and overall well-written paper. It will be a solid contribution to the Biosensors and will certainly appeal to many of its readers. I address some of the main issues with the manuscript in the next few paragraphs. It is recommended that this manuscript be published in Biosensors after completing revision.

1.      There are a lot of writing mistakes in the full text, please check and modify them, such as wrongly writing “SCJ” as “CSJ” in 1.1. And please adjust the position of Table 1, so that the whole table is in the same page.

2.      In 1.3, there is no picture. The text is not clear enough, please add some pictures for better understanding.

3.      Figure 2 does not correspond well to the description of the previous text. The previous text has a lot of description on the size of detection, but Figure 2 is not ieasily understood. Please replace it with a more suitable picture.

4.      In part 3, only 3.1 (callascope) has a data table to support the description of the clinical test. More data tables are needed to support clinical tests of other four detections.

5.      The whole text introduces five detections and their clinical tests for cervical cancer. But there is no comparison between them. Which one is more suitable for today's society?

6.      The introduction illustrates some treatments for cervical cancer. In order to better support this statement, the following recently published important related papers should be cited:  Adv Mater. 2022, 34, 2106388.

Author Response

We thank the reviewer for the thorough review of our paper. We have addressed each issue and concern individually.

  1. There are a lot of writing mistakes in the full text, please check and modify them, such as wrongly writing “SCJ” as “CSJ” in 1. And please adjust the position of Table 1, so that the whole table is in the same page.

RESPONSE: We thank the reviewer for the positive assessment of our work, and we apologize for the oversights and grammatical mistakes. We have gone through the paper with great care and believe this issue to be solved.

  1. In 3, there is no picture. The text is not clear enough, please add some pictures for better understanding.

RESPONSE: The authors thank the reviewer for the feedback. The text has been modified in order to appear clearer to the reader and should lead to the better understanding of the section (and manuscript overall).

  1. Figure 2does not correspond well to the description of the previous text. The previous text has a lot of description on the size of detection, but Figure 2 is not ieasily understood. Please replace it with a more suitable picture.

RESPONSE: Figure 2 shows the Calla Lilly shape of the inserter integrated with the Callascope as the text in section 3.1.1 describes. The figure is necessary since the subsection talks about the device and therefore an image of the device completes the story.

  1. In part 3, only 3.1 (callascope) has a data table to support the description of the clinical test. More data tables are needed to support clinical tests of other four detections.

RESPONSE: The table under section 3.1 was employed to show the cervical area visible for the Callascope in two pilot studies. This is the only device where visualization of the area is calculated and reported. The majority of the other devices report the sensitivity and specificity of the device, which is written out in paragraph form and the authors believe adding these to a table would be redundant. 

  1. The whole text introduces five detections and their clinical tests for cervical cancer. But there is no comparison between them. Which one is more suitable for today's society?

RESPONSE: Five cervical screening devices have been introduced along with a device description and clinical summary of existing data.  Devices with extensive clinical data report the sensitivity and specificity the original authors calculated as compared with commonly used methodology (e.g. a colposcope) and devices with less extensive clinical exposure summarized what the original authors concluded during the pilot studies. All these devices provide different images and quantitative information, as well as different fields of view, power consumption, etc. These differences have been summarized in Table 2 and each modality could potentially be useful in the low-resource setting. This has been addressed in the conclusions.

  1. The introduction illustrates some treatments for cervical cancer. In order to better support this statement, the following recently published important related papers should be cited:  Adv Mater.202234, 2106388.

RESPONSE: We have added the paper in the introduction.

Reviewer 2 Report

The reviewer has read the manuscript with great interest. Most of the part is organized perfectly. However, a few corrections and improvements are needed in the revised submission. These comments include:

[1]   As a review article, the '1.2 Disease progression' section's first paragraph must cite appropriate references.

[2]   Some reference is cited in the middle and somewhere at the end of the sentence. It should be uniform style.

[3]   In Figures 7 and 8, the authors need to add the meaning of the arrow symbol to the figure description.

[4]    It will be convenient if the conclusion is in one paragraph. The authors have to concise the conclusion part with a meaningful summary.

Author Response

We thank the reviewer for the thorough review of our paper. We have addressed each issue and concern individually.

[1]   As a review article, the '1.2 Disease progression' section's first paragraph must cite appropriate references.

RESPONSE: Per the reviewer’s suggestion, I have added three references (Novikova et al., 2017, Sahebali et al., 2010 and Safaeian et al., 2007).

[2]   Some reference is cited in the middle and somewhere at the end of the sentence. It should be uniform style.

RESPONSE: We have changed the reference position to make it more uniform.

[3]   In Figures 7 and 8, the authors need to add the meaning of the arrow symbol to the figure description.

RESPONSE: We have added a description to the figure.

[4]    It will be convenient if the conclusion is in one paragraph. The authors have to concise the conclusion part with a meaningful summary.

RESPONSE: The authors thank the reviewer for the suggestion. The conclusion has been separated into two paragraphs, where the first paragraph summarizes the information related to the manuscript and the second paragraph shows the limitations and where future research can be focused. The authors believe this is the most straightforward way to show the reader a summary and conclusion of the work.